# Synthesis, Characterization, and Modification of Alumina Nanoparticles for Cationic Dye Removal

**DOI:** 10.3390/ma12030450

**Published:** 2019-02-01

**Authors:** Thi Phuong Minh Chu, Ngoc Trung Nguyen, Thi Lan Vu, Thi Huong Dao, Lan Chi Dinh, Hai Long Nguyen, Thu Ha Hoang, Thanh Son Le, Tien Duc Pham

**Affiliations:** 1Faculty of Chemistry, VNU-University of Science, Vietnam National University, Hanoi, 19 Le Thanh Tong, Hoan Kiem, Hanoi 10000, Vietnam; minhxndpvp@gmail.com (T.P.M.C.); trungnn.hus@gmail.com (N.T.N.); vuthilan_t59@hus.edu.vn (T.L.V.); daohuongk56a1993@gmail.com (T.H.D.); 2HUS High School for Gifted Students, VNU-University of Science, Vietnam National University, Hanoi, 182 Luong The Vinh, Thanh Xuan, Hanoi 10000, Vietnam; dinhchi55@gmail.com (L.C.D.); ceonguyenhailong@gmail.com (H.L.N.); 3High School of Education Sciences, University of Education, Vietnam National University, Hanoi, Kieu Mai, Phuc Dien, Bac Tu Liem, Hanoi 10000, Vietnam; hoangthuha0105@yahoo.com

**Keywords:** Alumina nanoparticles, Rhodamine B, SDS, Adsorption isotherm, Two-step model

## Abstract

In the present study, alumina nanoparticles (nano-alumina) which were successfully fabricated by solvothermal method, were characterized by X-ray diffraction (XRD), Fourier transform infrared spectroscopy (FT-IR), Transmission Electron Microscopy (TEM), and Brunauer–Emmett–Teller (BET) methods. The removal of cationic dye, Rhodamine B (RhB), through adsorption method using synthesized nano-alumina with surface modification by anionic surfactant was also investigated. An anionic surfactant, sodium dodecyl sulfate (SDS) was used to modify nano-alumina surface at low pH and high ionic strength increased the removal efficiency of RhB significantly. The optimum adsorption conditions of contact time, pH, and adsorbent dosage for RhB removal using SDS modified nano-alumina (SMNA) were found to be 120 min, pH 4, and 5 mg/mL respectively. The RhB removal using SMNA reached a very high removal efficiency of 100%. After four times regeneration of adsorbent, the removal efficiency of RhB using SMNA was still higher than 86%. Adsorption isotherms of RhB onto SMNA at different salt concentrations were fitted well by a two-step model. A very high adsorption capacity of RhB onto SMNA of 165 mg/g was achieved. Adsorption mechanisms of RhB onto SMNA were discussed on the basis of the changes in surface modifications, the change in surface charges and adsorption isotherms.

## 1. Introduction

Removal of organic dye from a water environment is important in environmental remediation. Organic dye is one kind of popular pollutant that is can released from many industrial activities related to cosmetic, paint, pigments, textile, paper, etc. [1]. Almost charged organic dyes (ionic dye) are colored and highly toxic [2]. Nowadays, numerous techniques have been used ionic dye treatment like adsorption [3,4,5,6], degradation using photocatalyst [7,8], electrochemical oxidation [9,10], coagulation/flocculation [11], and biodegradation [12]. Among them, adsorption is one of the most effective methods for ionic dye treatment. Adsorption technique also is suitable for the developing countries by using low cost adsorbents [13,14]. Metal oxides are main components of natural soil that are a cheap adsorbent.

Alumina is a well-known metal oxide material that is one of the most common adsorbents used in environmental engineering. Alumina has numerous structural phases, namely α, β, γ, η, θ, κ, and χ [15]. It is found that gamma alumina (γ-Al_2_O_3_) has high specific surface area, especially nano γ-Al_2_O_3_, which is powerful enough to be an effective adsorbent for ionic dye. Nevertheless, γ-Al_2_O_3_ has low negative charge density in the neutral pH [16]. Thus, it is hard to remove cationic dye due to small electrostatic attraction between cationic dye and negatively charged alumina surface. In this case, surface modification of alumina is necessary. Many studies focused on the modification of alumina by using ionic surfactants to enhance removal efficiency of both organic and inorganic pollutants [1,17,18,19,20]. Rhodamine B (RhB), which is a cationic dye, is widely used in many industrial activities. Rhodamine B is also well known to occur in wastewater [21,22,23]. To our best knowledge, the adsorptive removal of RhB using surfactant modified synthesized alumina nanoparticles has not been reported.

In order to understand adsorption process systematically, isothermal condition is basically evaluated by modeling. Langmuir and Freundlich isotherms are the most common models for adsorption [24,25]. Nevertheless, these isotherms are not applicable for surfactant adsorption. Interestingly, a two-step adsorption derived by Zhu et al. [26] successfully described various adsorption systems of surfactants, polymers, and antibiotics [6,26,27,28,29,30,31,32]. Thus, this model could be suitable for RhB adsorption onto surfactant modified alumina.

The aim of this work is to investigate the removal of RhB by adsorption technique using sodium dodecyl sulfate (SDS) modified nano γ-Al_2_O_3_ (SMNA) after surface modification of γ-Al_2_O_3_ with SDS solution. Some effective parameters for RhB removal using after characterization of γ-Al_2_O_3_ by X-ray diffraction (XRD), Fourier transform infrared spectroscopy (FT-IR), Transmission Electron Microscopy (TEM), and Brunauer–Emmett–Teller (BET) methods were also studied. The adsorption isotherms of RhB adsorption onto SMNA were investigated by both experimental and modeling. Adsorption mechanisms of RhB onto SMA are discussed on the basis of adsorption isotherms, the surface charge changes of nano γ-Al_2_O_3_ after adsorption by ζ potential and the changes of surface functional groups by Fourier transform infrared spectroscopy (FT-IR), respectively. The regeneration of SMNA was carried out in the present study to evaluate the reuse adsorbent.

## 2. Materials and Methods

### 2.1. Materials 

Aluminum nitrate Al(NO_3_)_3_·9H_2_O and NaOH pellets which were pure analytical reagents, were purchased from Merck, Darmstadt, Germany. 

Cationic dye, Rhodamine B (CAS 81-88-9) for microscopy (with purity > 95.0%), with a molecular weight of 479.02 g/mol, was delivered from Merck (Darmstadt, Germany). An anionic surfactant, sodium dodecyl sulfate (SDS) (with purity greater 95%) supplied from Scharlau (Barcelona, Spain, EU), was used without further purification to modify synthesized nano-alumina. Figure 1 shows the chemical structures of RhB (A) and SDS (B). The cationic dye, methylene blue (with purity > 98.5%) and organic solvent chloroform CHCl_3_ (HPLC grade) from Merck (Darmstadt, Germany) were used to quantify the concentrations of SDS. Ionic strength and pH were adjusted by the addition of NaCl (p.A, Merck, Darmstadt, Germany), HCl and NaOH (volumetric analysis grade, Merck, Darmstadt, Germany). Solution pH was conducted using an HI 2215 pH meter (Hanna, Woonsocket, RI, USA). The glass pH electrode was checked for calibration with three standard buffers of 4.01, 7.01, and 10.01 (Hanna, Woonsocket, RI, USA). Other chemicals with analytical grade were purchased from Merck (Darmstadt, Germany). Ultrapure water was taken from ultrapure water system (Labconco, Kansai, MO, USA) with resistivity 18.2 MΩ·cm was used in preparing all aqueous solution.

### 2.2. Fabrication of Alumina Nanoparticles

Alumina nanoparticles were fabricated according to previous study with a modification [33]. The solutions of Al(NO_3_)_3_ and NaOH were used to synthesize alumina. The 4M NaOH solution was prepared by dissolving 12.2449 g NaOH pellets in 75.0 mL ultrapure water. A solution of 1 M Al(NO_3_)_3_ was prepared by dissolving 37.50 g of Al(NO_3_)_3_·9H_2_O in 100.0 mL ultrapure water. Aluminum hydroxide was obtained by slowly adding 1 M Al(NO_3_)_3_ with 4 M NaOH in a plastic vessel. White precipitation of aluminum hydroxide formed from this state was then separated by using a centrifuge at 6000 rpm (Digisytem, Taiwan). After that, the samples were dried in thermal oven at 80 °C for 24 h. The obtained powder was then calcined at 600 °C for 12h in thermal furnace before cooling to room temperature in a dessicator. Finally, alumina particles were stored in a polyethylene container.

### 2.3. Characterization Methods

The synthesized alumina was characterized by X-ray diffraction (XRD), Fourier transform infrared spectroscopy (FT-IR), Transmission Electron Microscopy (TEM), Brunauer–Emmett–Teller (BET) methods and ζ potential measurements.

The XRD pattern was performed on a Bruker D8 Advance X-ray Diffractometer (Karlsruhe, German) with CuK*_α_* radiation (*λ* = 1.5418 Å). The XRD pattern was recorded in a range of 20–80° (2θ) with a step size of 0.03°. The FT-IR spectra was collected with an Affinity-1S spectrometer (Shimadzu, Kyoto, Japan). The FTIR spectra of nano-alumina particles, SDS modified nano-alumina (SMNA) at the plateau adsorption level, SMNA after RhB adsorption and RhB salt were obtained with a resolution of 4 cm^−1^ at atmospheric pressure (25 °C).

BET method was used using a surface area and pore size Analyzer surface area analyzer Micromerities (TriStar 3000, Norcross, GA, USA). The adsorption isotherm of nitrogen (N_2_) was conducted in a 9 mL cell with outgas condition of 150 °C in 90 min. The particle size distribution of synthesized nano-alumina which was evaluated by TEM, was performed by using Hitachi (H7650, Tokyo, Japan) with Olympus camera (Veleta 2000 × 2000). The operating conditions were accelerating condition of 80 kV and exposure time of 1 s.

The surface charge of synthesized nano-alumina, SMNA and SMNA after RhB adsorption in 1 mM NaCl, at pH 4 were examined by using Zetasizer Nano ZS (Malvern, Worcestershire, UK). Dynamic Light Scattering (DLS) for particle size characterization in the solutions of nano-alumina and SMNA was also conducted with Zetasizer Nano ZS by applying backscattering detection (173° detection optics) at 22 °C. A cleaned plastic cuvette containing 2 mL of the sample was used in each DLS measurement.

The zeta (ζ) potential was calculated from electrophoretic mobility using Smoluchowski’s equation [34].
(1)$$\zeta =\frac{u_{e}\eta }{\varepsilon_{rs}\varepsilon_{0}}$$
where ζ is the ζ potential (mV), *u_e_* is the electrophoretic mobility (µm cm/sV), *η* is the dynamic viscosity of the liquid (mPa·s), *ε_rs_* is the relative permittivity constant of the electrolyte solution, and *ε*_0_ is the electric permittivity of the vacuum (8.854 × 10^−12^ F/m).

### 2.4. Adsorption Studies

All adsorption experiments were conducted by batches in 15mL Falcon tubes at 25 ± 2 °C controlled by an air conditioner. All adsorption of SDS and RhB onto synthesized nano-alumina were carried out in triplicates.

For SDS adsorption onto synthesized nano-alumina, a fixed 50 mg/mL of synthesized nano-alumina was mixed with 10 mL of different SDS concentrations at 10 and 100 mM NaCl for 2 h to form SDS modified nano-alumina (SMNA). The concentrations of SDS were quantified by an extractive spectrophotometric method using the ion paired formation complex of SDS and methylene blue in chloroform solvent. Detail of the procedure was published in our previously published paper [30].

For RhB adsorption, a known amount of adsorbent was thoroughly mixed with 10 mL aqueous RhB in different NaCl concentration. The effective conditions (pH, adsorption time, adsorbent dosage, ionic strength) on adsorption RhB were studied. The concentrations of RhB were determined by using Ultraviolet Visible (UV-Vis) spectroscopy at a wavelength of 554 nm with a quartz cuvette with a 1 cm optical path length using a spectrophotometer (UV-1650 PC, Shimadzu, Kyoto, Japan). The molar absorbance coefficient of RhB determined by experiment is 105,456 ± 734 cm^−1^·M^−1^ that is very good agreement with the value of 106,000 cm^−1^·M^−1^ for standard RhB.

The adsorption capacities of SDS onto nano-alumina and RhB onto SMNA were determined by the following equation.
(2)$$\Gamma =\frac{Ci-\;Ce}{m}\;\times M\times 1000$$
where *Γ* is the adsorption capacity of SDS or RhB (mg/g), C*i* is the initial concentration of SDS or RhB (mol/L), *C_e_* is the equilibrium concentration of SDS or RhB (mol/L), *M* is molecular weight of SDS or RhB (g/mol), and *m* is the adsorbent dosage (mg/mL). 

The removal efficiency (%) of RhB was calculated by Equation (3).
(3)$$\mathrm{Removal}\;\mathrm{efficiency}\;(\%)\;=\;\frac{C_{i}-C_{e}}{C_{i}}\times 100\%$$
where *C_i_* and C*_e_* are initial concentration and equilibrium concentration of RhB (mol/L), respectively. 

The obtained isotherms were fitted by general isotherm equation that could be applied to describe adsorption isotherms of RhB onto SMNA. The general isotherm equation [26] is:(4)$$\Gamma =\frac{\Gamma_{\infty }k_{1}C\left(\frac{1}{n}+k_{2}C^{n-1}\right)}{1+k_{1}C\left(1+k_{2}C^{n-1}\right)}$$where Γ is the amount of adsorbed RhB at concentration C, $\Gamma_{\infty }$ is the maximum adsorption at high concentrations, $k_{1}$ and $k_{2}$ are equilibrium constants involved in the first and second step, respectively, and n is clusters of multilayer. C is the equilibrium concentrations of RhB.

## 3. Results and Discussion 

### 3.1. Characterization of Synthesized Alumina Nanoparticles

The XRD pattern of the alumina nanoparticles obtained by the solvothermal method is shown in Figure 2. The sharp peaks with high intensity at 2θ = 38° and 67° indicate the high crystalline of alumina. The gamma phase of alumina was confirmed due to the presence of the peaks at 46° and 61° [35].

The FT-IR spectra of the nano-alumina shown in Figure 3 indicated that the peaks appearing at 3657.04, 3550.95, and 3622.32 cm^−1^ are assigned for –OH stretching vibration in aluminum hydr(oxide) structure. The peaks appeared at 1031.92, 520.78, and 426.27 cm^−1^, corresponding to Al-O bending vibration of Al-OH group [36]. The broader peaks between 1000 cm^−1^, 981.77 cm^−1^, and 520.78 cm^−1^ confirmed the bending vibration of Al-O bond [35]. 

Transmission electron microscopy (TEM) was performed to characterize the morphology and particle size of alumina powers (Figure 4). The TEM image shown in Figure 4 shows that alumina particles are sphered ones with the size in the range of 30–40 nm, indicating that synthesized alumina is a nanosized powder. The DLS data (not shown in detail) indicates that the hydrodynamic Z-average diameter of nano-alumina was about 410–450nm at pH 4 that much higher than the particle size of nano-alumina due to the aggregation of nano-alumina in NaCl concentration. The Z-average diameter of SMNA was about 924–958 nm, indicating that the formation of SMNA aggregates in the presence of SDS. However, the aggregation of SMNA did not induce significantly to RhB adsorption.

The specific surface area of the nano γ-Al_2_O_3_ by BET was calculated from N_2_ adsorption isotherm and found to be around 221.3 m^2^/g (Figure 5).

The specific surface area in our work is similar to the results of alumina published by Khataee et al. [37]. It implies that the synthesized nano γ-Al_2_O_3_ has high specific surface area that is good for adsorptive removal of ionic dye. However, in order to enhance the removal efficiency of RhB using nano γ-Al_2_O_3_, the surface charge modification is needed because the electrostatic interaction is more important for ionic dye adsorption [1].

### 3.2. Modification of Synthesized Nano-Alumina by SDS Adsorption

The synthesized nano γ-Al_2_O_3_ was modified by SDS pre-adsorption of at two salt concentration at pH 4. At pH 4, the surface charge of γ-Al_2_O_3_ is highly positive due to the point of zero charge 8.5 of alumina [16]. At pH < 4, the dissolution of alumina could occur so that the characteristics may be changed [38]. Figure 6 shows that adsorption of SDS onto nano γ-Al_2_O_3_ grew up with an increase of salt from 10 to 100 mM. Adsorption of anionic SDS onto nano γ-Al_2_O_3_ here is different to adsorption of SDS onto activated beads of Al_2_O_3_ in which the electrostatic attraction is main driving force [39]. In this case, SDS adsorption onto nano γ-Al_2_O_3_ is controlled by both electrostatic and hydrophobic interactions. The maximum adsorption capacity was obtained at 10 mM NaCl when the SDS concentration was greater than 0.01 M. It should be noted that the micelles are surely formed in the solution with 0.01 M SDS because its concentration is much the higher than the critical micelle concentration (CMC). The admicelle of SDS molecules are completely formed on γ-Al_2_O_3_ surface that the charge reversal is occurred. The plateau adsorption capacity of SDS onto γ-Al_2_O_3_ reaches to 450 mg/g that is similar to the SDS adsorption onto alumina in the previous paper [40]. With the presence of a high number of admicelles on γ-Al_2_O_3_, the removal of cationic dye RhB using highly negative charge γ-Al_2_O_3_ particles can increase significantly. Therefore, the adsorption of initial 0.01 M SDS onto γ-Al_2_O_3_ was fixed to modify the γ-Al_2_O_3_ surface that was carried out at 100 mM NaCl (pH 4).

### 3.3. Adsorptive Removal of RhB Using Synthesized Nano-Alumina (NA) without and with SDS Modification

#### 3.3.1. Effect of pH

The removal of RhB using synthesized nano γ-Al_2_O_3_ is strongly influenced by pH because pH highly affects to the charging behavior of nano γ-Al_2_O_3_ and the desorption of SDS [30,32]. The effects of pH on adsorptive removal of RB using synthesized nano γ-Al_2_O_3_ without and with SDS modification were conducted in the pH range 3–10 in 1 mM NaCl with contact time of 180 min and adsorbent dosage of 5 mg/mL (Figure 7).

Figure 7 indicates that the removal efficiency decreased with increasing pH from 3 to 10 for synthesized nano γ-Al_2_O_3_ without while the removal reduced from pH 4 to 10. At pH 3, the dissolution of alumina is occurred so that the property may be changed [41]. When pH solution increases, the positive charge of nano γ-Al_2_O_3_ is decreased whereas RhB is cationic dye in such range of pH. For SDS modified nano γ-Al_2_O_3_ (SMNA), the SDS desorption may be enhanced so that the less negatively charge SDS modified alumina is occurred [32]. It should be noted that the initial concentration of 10^−4^ M RhB using SMNA is greater than that for the case without SDS (10^−6^ M). Nevertheless, the removal efficiency of RhB using SMNA is always higher than that using synthesized nano γ-Al_2_O_3_. It implies that the removal efficiency increased significantly using modified nano γ-Al_2_O_3_ (SMNA) compared without SDS modification. Maximum removal efficiency achieved 97.7% at pH 4 for SMNA. At the same RhB concentration of 10^−4^ M, the removal efficiency of RhB using alumina is smaller than 40%. Therefore, pH 4 is chosen for further adsorption study of RhB onto two adsorbents.

#### 3.3.2. Effect of Adsorption Time

Adsorption time affects the completeness of adsorption equilibration. The effect of contact time on the adsorptive removal of RhB using synthesized nano γ-Al_2_O_3_ with and without SDS from 10 to 240 min is presented in Figure 8. Figure 8 shows that the adsorption reaches equilibrium 90 min and 180 min for SMNA and without SDS modification, respectively. The adsorption time of RhB onto nano γ-Al_2_O_3_ with SDS modification is much faster than that without SDS. It is also faster than RhB adsorption on traditional activated carbon (120 min) [42]. The adsorption time of 90 min is fixed for RhB adsorption onto SMNA and 180 min is selected for one onto synthesized nano γ-Al_2_O_3_.

#### 3.3.3. The Effect of Adsorbent Dosage

The adsorbent dosage highly affects to the adsorption process because it can induce the total specific surface area of adsorbent and number of binding sites [43]. The amounts of synthesized nano γ-Al_2_O_3_ with and without SDS were varied from 0.5 to 30 mg /mL (Figure 9). As can be seen in Figure 9, the removal of RhB using synthesized nano γ-Al_2_O_3_ without SDS increased with increasing adsorbent dosage, while SDS modified nano γ-Al_2_O_3_ required much smaller amount. When using SMNA, the amount of adsorbent is economical comparing with nano γ-Al_2_O_3_ without SDS. For removal of RhB using SMNA, the adsorbent dosage 5 mg/mL is good enough to achieve the efficiency of approximately 100%. Thus, optimum adsorbent dosage is found to be 5 mg/mL.

Because the removal efficiency of RhB using SMNA is much higher than nano γ-Al_2_O_3_ without SDS, we only focus on the adsorption mechanisms of RhB onto SMNA in the next section.

### 3.4. Adsorption Isotherms of RhB on SDS Modified Nano-Alumina (SMNA)

The effect of ionic strength on adsorption of RhB onto SDS modified nano γ-Al_2_O_3_ (SMNA) is clearly observed on the isotherms (Figure 10). At pH 4, the adsorption of RhB decreases with increasing NaCl concentration at low RhB concentration. Nevertheless, at high RhB concentration adsorption increases with an increase of ionic strength. At low RhB adsorption, an increase in salt concentration increased the number of cation Na^+^ on the negatively charge layer of SMNA, decreasing the electrostatic effect of RhB on negatively charged SMNA. The electrostatic attraction between the positive charge of RhB and negatively charge of alumina is significantly screened by increasing salt concentrations. However, other interactions such as surface complexation and Van der Waals interactions and hydrogen bonding can induce adsorption at high RhB concentration. It is suggested that adsorption of RhB onto SMNA is controlled by both electrostatic and non-electrostatic interactions. The isotherms of RhB show a common intersection point (CIP) in which the electrostatic contribution to the adsorption vanishes and the salt effect disappears [30]. 

Figure 10 also shows that at different NaCl concentrations, the experimental results of RhB onto SMNA can be represented well by general isothermal equation, Equation (4), with the fitting parameters in Table 1. At different salt concentrations, the maximum adsorption capacity of RhB at 100 mM is higher than that at 1 mM. Table 1 also shows that we can use the same fitting parameters (*k*_2_ and *n*) for isotherms at 1 and 100 mM NaCl. However, the value of *k*_1_ at 1 mM NaCl is 10 times greater than that at 100 mM. It suggests that *k*_1_ is a valuable parameter to evaluate electrostatic interaction at low RhB concentration. The higher the salt concentration, the higher value *k*_1_ is obtained.

### 3.5. Adsorption Mechanisms of RhB onto SDS Modified Nano γ-Al_2_O_3_ (SMNA)

In this section, adsorption mechanisms of RhB onto SDS modified nano γ-Al_2_O_3_ is discussed in detail based on the change in surface charge by monitoring ζ potential and the change in functional groups by FT-IR and adsorption isotherms of RhB onto SMNA.

The ζ potential in Figure 11 shows that the ζ potential of synthesized nano γ-Al_2_O_3_ is highly positive at pH 4 (ζ = +47.55 mV). After surface modification with SDS to form SDS modified nano γ-Al_2_O_3_ (SMNA), the surface charge of adsorbent changed from positive to negative (ζ = −13.95 mV). It implies that the admicelles were occurred onto nano γ-Al_2_O_3_ surface and the charge was taken place in the presence of anion DS^−^ [30,32]. Nevertheless, after adsorption of cationic dye RhB, the negative charge of SMNA changed from negative to slight positive (ζ = +2.85 mV). The results of ζ potential suggest that RhB adsorption onto SMNA is due to electrostatic interaction.

FTIR is powerful tool to evaluate the change in functional groups after adsorption [44]. Figure 12 shows the FT-IR spectrum of the FT-IR spectrum of SMNA (A) and SMNA after RhB adsorption (B) in the wavenumber range 400–4000 cm^−1^.

We can see that the strong band of stretching of –OH appear in the wavelength of about 3658.96 cm^−1^ for SMNA (Figure 12A) is similar to FT-IR spectrum of synthesized nano γ-Al_2_O_3_ (Figure 3). Figure 12A also shows that FT-IR spectra of alumina after SDS adsorption are similar to the raw one (Figure 3). In additive, the relative intensity of asymmetrical and symmetrical stretching of –CH_2_–presented at 2920.23 and 2850.79 cm^−1^ appeared with very high intensity in FT-IR spectra of SMNA [45]. This confirms that the hydrophobic interaction occurred on the surface of alumina. In addition, the characteristics of SO_4_^2−^ at about 1226.73 cm^−1^ appear very strongly in spectra of SDS while all bands disappear in the spectra of alumina [39]. It suggests that SDS has a sulfate head group in contact with the surface of alumina nanoparticles via electrostatic attraction at high salt concentration (100 mM NaCl). In other words, the modification of alumina was successful due to the presence of bilayer or admicelles, or both, on the surface of alumina. 

On the one hand, the peak of –CH_2_– presented at 2061.90 cm^−1^ of SMNA disappears in the spectra of SMNA after RhB adsorption. The peaks in a range of 1591.27 cm^−1^ to 1651.07 cm^−1^ of RhB characterizing for N–H bonds [46] appeared on the surface of SMNA. The FT-IR spectra of SMNA after RhB adsorption suggest that RhB mainly adsorb the surface of Al_2_O_3_ by both electrostatic attractions as well as hydrophobic interaction. These results are in good agreement with RhB adsorption isotherm in which RhB adsorption onto SMNA is controlled by both electrostatic and non-electrostatic interactions.

### 3.6. Comparison of Effectiveness of Surfactant-Modified Nano γ-Al_2_O_3_ (SMNA) and Other Adsorbents for RhB Removal and Other Nano γ-Al_2_O_3_

The synthesized nano γ-Al_2_O_3_ is a novel adsorbent for RhB removal after surface modification with SDS. The removal efficiency is about 100% and very high adsorption capacity of 165 mg/g is achieved at the optimum adsorption conditions. We found that various scientific papers reported the removal of RhB using many kinds of adsorbents. To our best knowledge, the removal of RhB using SMNA has not been investigated. Furthermore, the SMNA used in the present study achieved highest adsorption capacity and absolute removal efficiency compared to other adsorbents (Table 2). 

In order to emphasize the high performance of SMNA for dye removal, let us discuss the potential in term of removal of cationic dye, methylene blue (MTB). The previously published paper by Ali el al. [51] successfully synthesized nano γ-Al_2_O_3_ by a precipitant method in the presence of nonionic surfactant Tween-80. This procedure required the ultrasonication and washing with ethanol before calcination at 550 °C. In our case, the nano γ-Al_2_O_3_ was easily synthesized by titrating NaOH into Al(NO_3_)_3_ without any other chemicals. The nano γ-Al_2_O_3_ in this study achieved better morphology and much higher specific surface area than that in the paper [51]. For cationic dye adsorption, MTB was removed with removal efficiency of 98% while the removal efficiency of RhB using SMNA in this paper reached to 100% for the similar initial concentration of cationic dye. It implies that nano γ-Al_2_O_3_ is an excellent adsorbent for MTB removal while SMNA is a novel adsorbent for RhB removal. 

For removal of pollutants by adsorption technique, the regeneration cycles of adsorbent is necessary to evaluate reuse potential and stability of SMNA adsorbent. Figure 13 shows the removal efficiency of OTC using SMA after four times of regeneration. A small decrease in removal efficiency is observed but it is insignificant. The RhB removal efficiency is still about 87% after four reused times. The error bars show that the standard deviations after four reused time experiments are very small, demonstrating that SMNA is not only a novel adsorbent but also applicable for regeneration. 

## 4. Conclusions

We successfully synthesized γ-Al_2_O_3_ nanoparticles and modified the surface of γ-Al_2_O_3_ by adsorption of anionic surfactant sodium dodecyl sulfate (SDS). The nano γ-Al_2_O_3_ has a mean particle size distribution of 40 nm while the functional surface groups of nano γ-Al_2_O_3_ was confirmed by Fourier transform infrared spectroscopy (FT-IR). A high specific surface area of nano γ-Al_2_O_3_ was found to by 221.3 m^2^/g. The SDS adsorption onto nano γ-Al_2_O_3_ was done at 100 mM NaCl forming bilayer of admicelles to enhance the removal of Rhodamine B dye. Under optimum adsorption conditions of RhB onto SDS modified nano-alumina (SMNA) including contact time 120 min, pH 4, and adsorbent dosage of 5 mg/mL, the removal efficiency reached 100% and adsorption capacity was 165.0 mg/g. After reuse four times, the removal efficiency of RhB using SMNA was higher than 86%. At different NaCl concentration, adsorption isotherms of RhB onto SMNA were in good agreement with the two-step model. Adsorption mechanisms of RhB onto SMNA were electrostatic attraction between cationic dye molecules and negatively charged SMNA surface at low RhB concentrations, while hydrophobic interaction was important controlled adsorption at high RhB concentrations. We indicate that SMNA is a smart nanomaterial for RhB removal from water environment.

## Figures and Tables

**Figure 1 materials-12-00450-f001:**
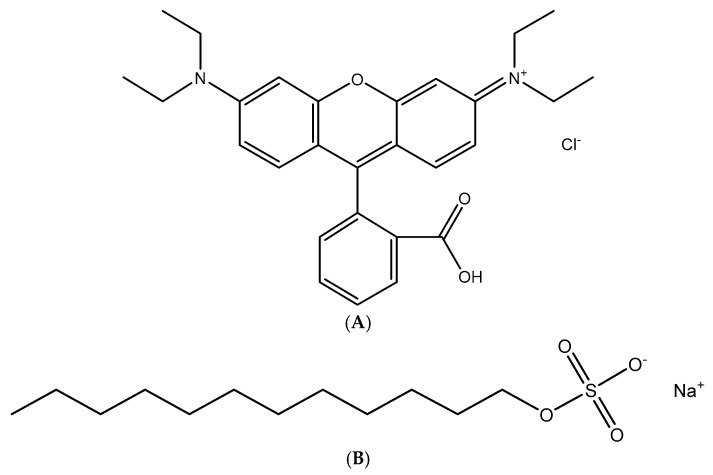
Chemical structures of Rhodamine B (RhB) (**A**) and sodium dodecyl sulfate (SDS) (**B**).

**Figure 2 materials-12-00450-f002:**
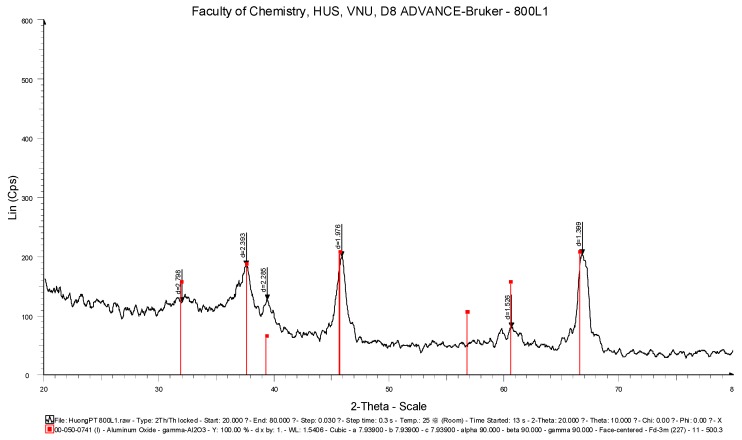
XRD pattern of synthesized γ-Al_2_O_3_ nanoparticles.

**Figure 3 materials-12-00450-f003:**
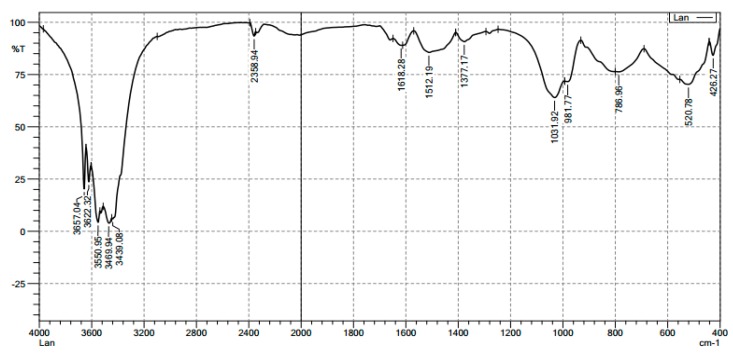
The FT-IR spectra of synthesized γ-Al_2_O_3_ nanoparticles.

**Figure 4 materials-12-00450-f004:**
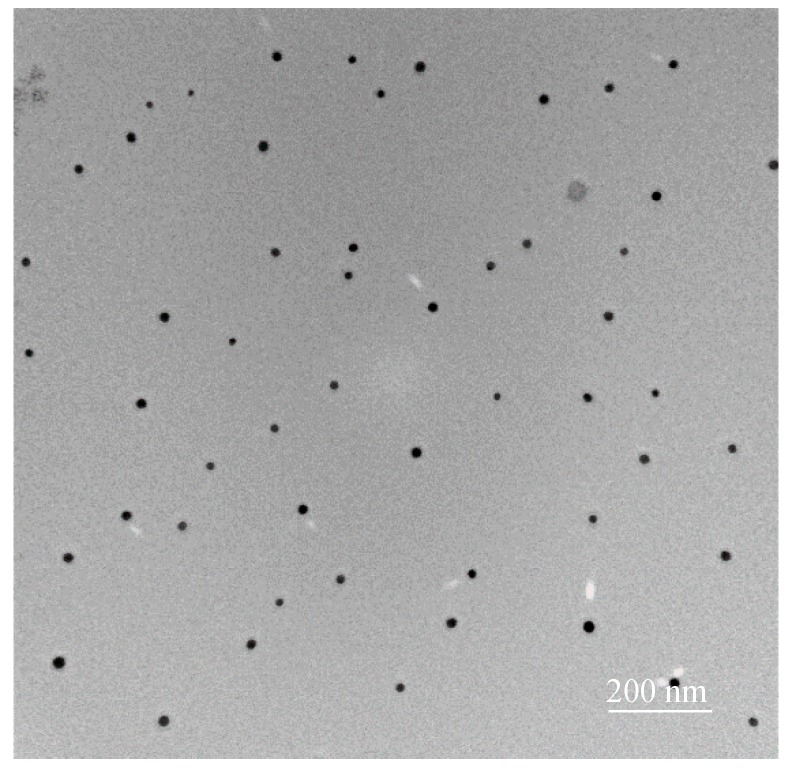
TEM image of synthesized γ-Al_2_O_3_ nanoparticles.

**Figure 5 materials-12-00450-f005:**
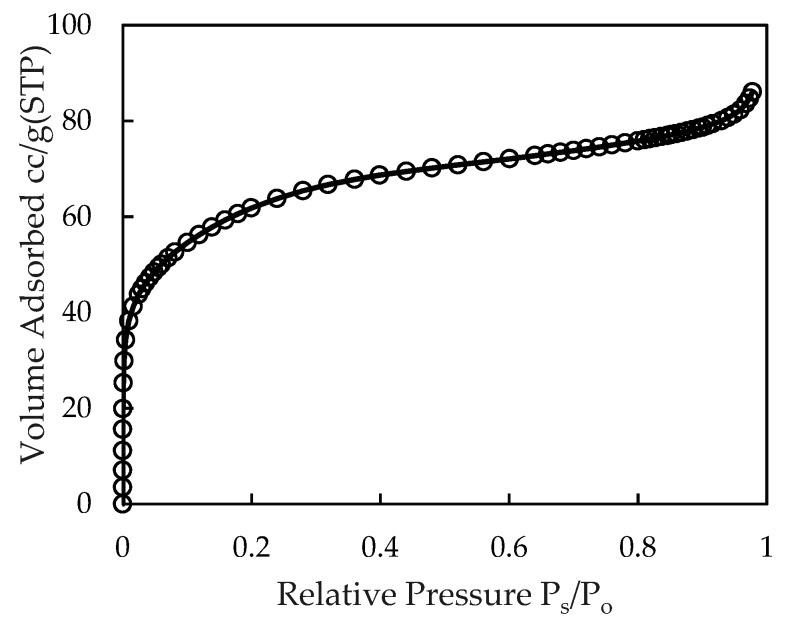
Adsorption isotherm of N_2_ onto synthesized nano-alumina.

**Figure 6 materials-12-00450-f006:**
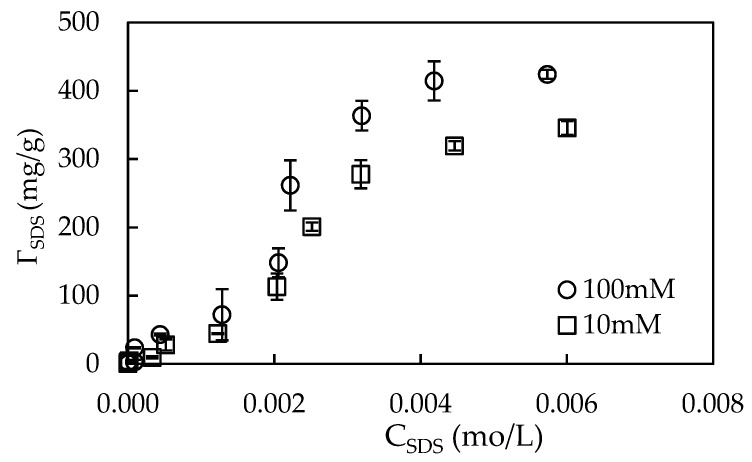
Adsorption of SDS onto synthesized nano-alumina at different NaCl concentrations.

**Figure 7 materials-12-00450-f007:**
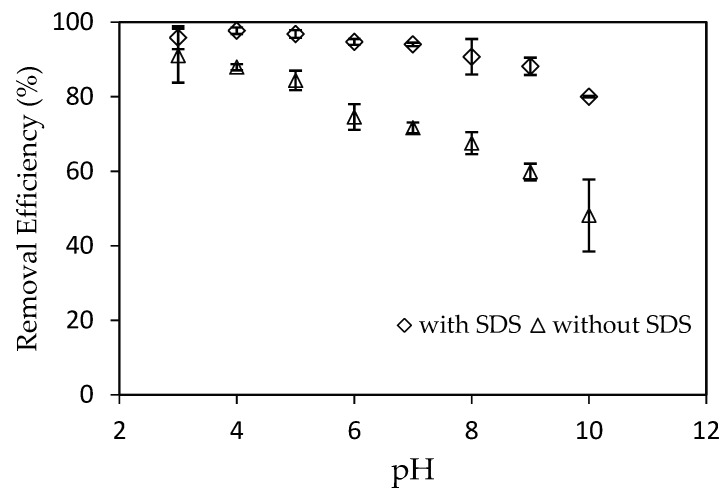
The effect of pH on RhB removal using synthesized nano γ-Al_2_O_3_ without SDS modification (C*i* (RhB) = 10^−6^ M and with SDS modification (C*i* (RhB) = 10^−4^ M).

**Figure 8 materials-12-00450-f008:**
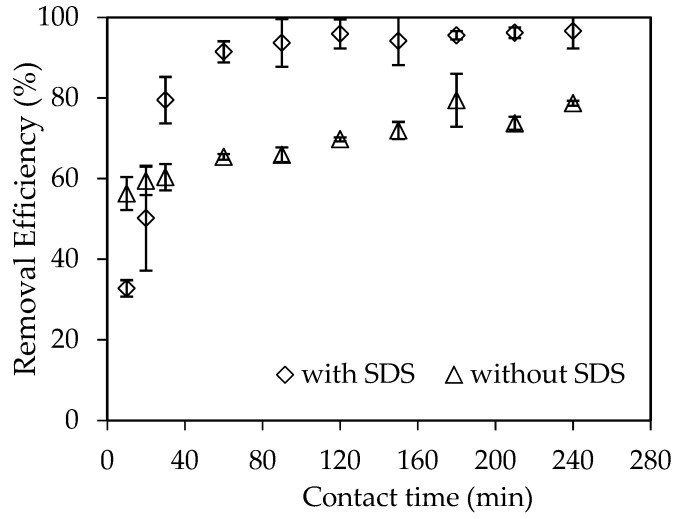
The effect of contact time on RhB removal using synthesized nano γ-Al_2_O_3_ without SDS modification (Temperature 25 ± 2 °C, *C_i_* (RhB) = 10^−6^ M and with SDS modification (*C_i_* (RhB) = 10^−4^ M).

**Figure 9 materials-12-00450-f009:**
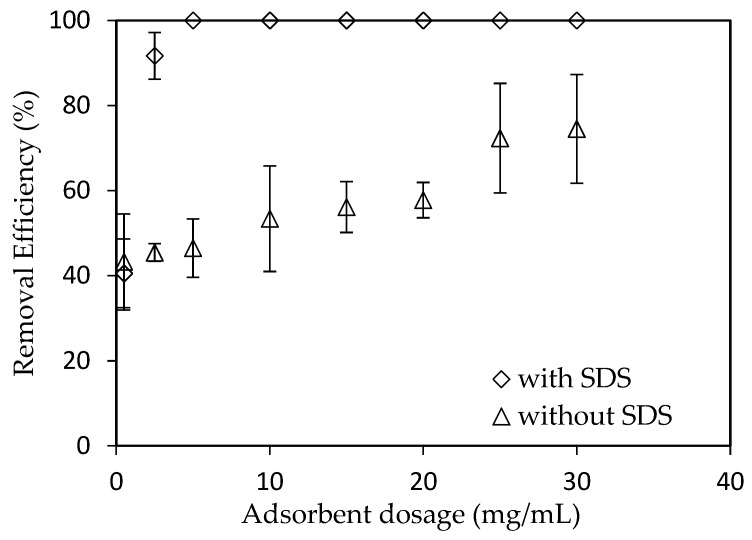
The effect of adsorbent dosage on RhB removal using synthesized nano γ-Al_2_O_3_ without SDS modification (*C_i_* (RhB) = 10^−6^ M) and with SDS modification. (*C_i_* (RhB) = 10^−4^ M).

**Figure 10 materials-12-00450-f010:**
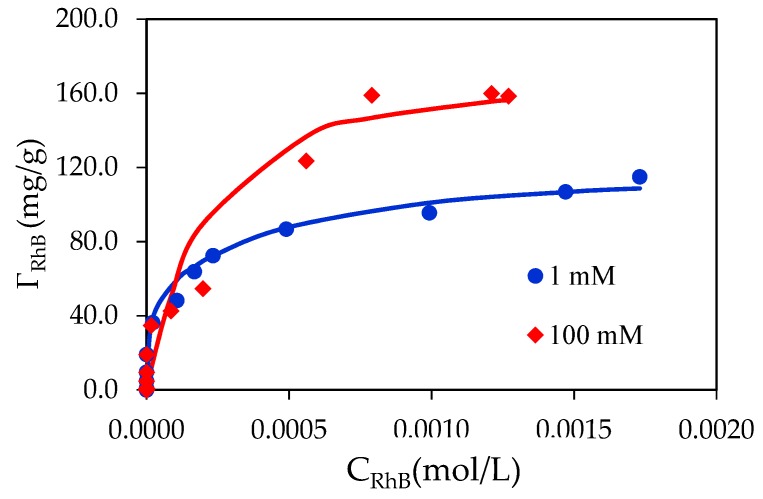
Adsorption isotherms of RhB onto SDS modified nano γ-Al_2_O_3_ (SMNA) at different NaCl concentrations. The points are experimental data while solid lines are fitted by a two-step adsorption model.

**Figure 11 materials-12-00450-f011:**
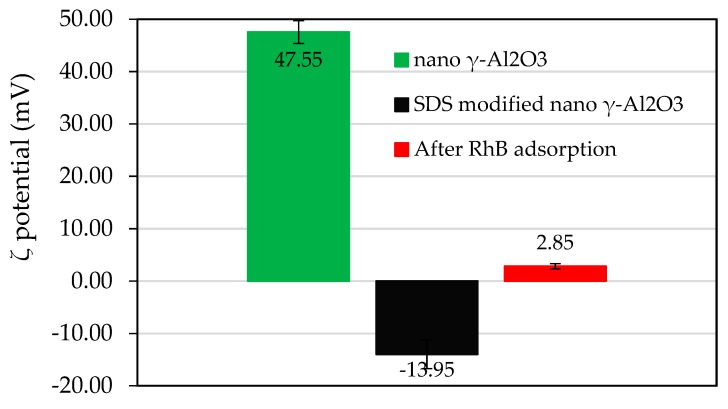
The ζ potential of synthesized nano γ-Al_2_O_3_, SDS modified nano γ-Al_2_O_3_ (SMNA), and SMNA after RhB adsorption in 1 mM NaCl (pH 4).

**Figure 12 materials-12-00450-f012:**
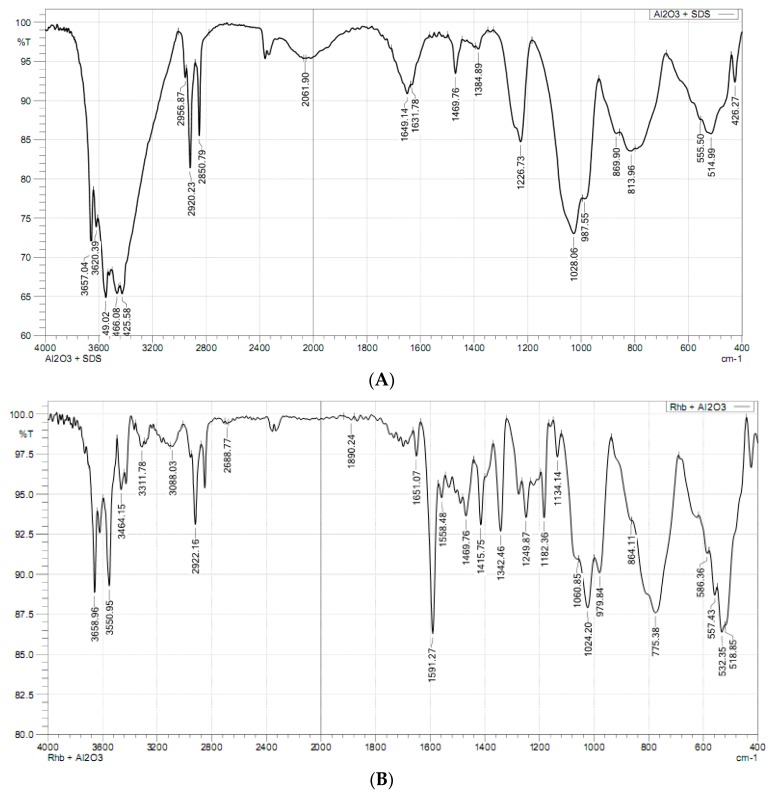
The FT-IR spectra of SDS modified nano γ-Al_2_O_3_ (**A**) and SMNA after RhB adsorption (**B**).

**Figure 13 materials-12-00450-f013:**
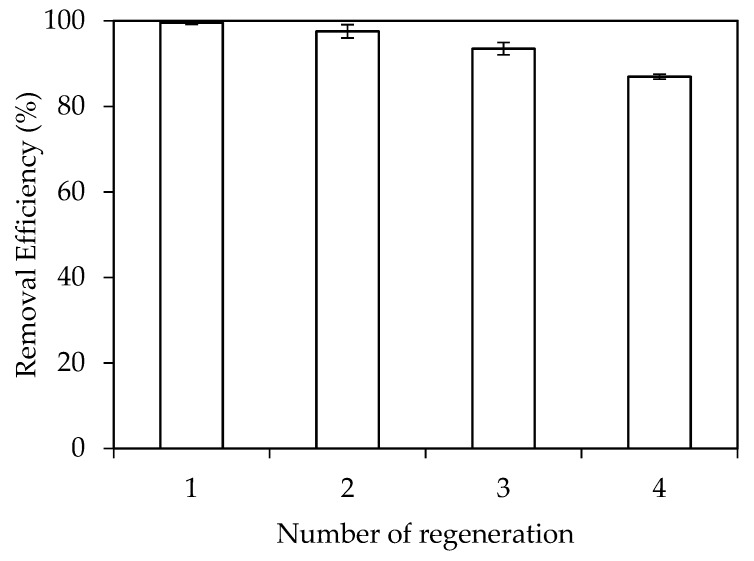
Removal efficiency of RhB using SMNA after four regenerations. Error bars show standard deviation of three replicates.

**Table 1 materials-12-00450-t001:** The fit parameters for adsorption RhB onto SDS modified nano γ-Al_2_O_3_ (SMNA).

C_NaCl_ (mM)	*Γ* (mg/g)	k_1_ (10^4^ g/mg)	k_2_ (10^3^ g/mg)^n−1^	n
100	165	10	20	2.2
1	120	100	20	2.2

**Table 2 materials-12-00450-t002:** Adsorption capacity and removal efficiency of surfactant-modified nano γ-Al_2_O_3_ (SMNA) and other absorbents for removal of Rhodamine B (RhB).

Adsorbent	Adsorption Capacity (mg/g)	Removal Efficiency (%)	References
Monodispersed mesoporous nanosilica	23.0	96	[47]
Polyamide grafted carbon microspheres	19.9	100	[46]
Polymeric nanotubes	35.58	99	[23]
Kaolinite	46.1	83	[48]
Sago waste activated carbon	16.1	100	[42]
Hypercross linked polymeric adsorbent	25.0	97	[49]
Surfactant-modified coconut coir pith	14.9	97	[50]
Surfactant-modified nano γ-Al_2_O_3_ (SMNA)	165.0	100	This study

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
