# Peer review of "Synthesis, Characterization, and Modification of Alumina Nanoparticles for Cationic Dye Removal"

_materials, 2019, doi:10.3390/ma12030450_

Reviewer 1 Report

Journal: Materials

Manuscript ID: materials-425828

Title: Synthesis, characterization and modification of alumina nanoparticles for cationic dye removal

Authors: Thi Phuong Minh Chu, Ngoc Trung Nguyen, Thi Lan Vu, Thi Huong Dao, Lan Chi Dinh, Hai Long Nguyen, Thu Ha Hoang, Thanh Son Le *, Tien Duc Pham *

Article Type: Article

The authors have reported preparation of anionic sodium dodecyl sulfate (SDS) modified nano-alumina (SMNA) and its adsorption property of Rhodamine B (RhB). Nano-alumina was characterized using X-ray diffraction, Fourier transform infrared spectroscopy, Transmission Electron Microscopy, and BET methods. The optimum adsorption conditions were studied. The experiments were well organized, and the manuscript is well written. Although I believe there is a merit for readers and publisher to publish of the manuscript in Materials. I feel that some parts should be improved. Please consider the following comments:

1. Line 76: Please indicate purity of RhB.

2. Line 80: Methylene blue was used in this study. However one cannot find out the results of methylene blue.

3. Line 116: Please indicate what kind of measurement. Maybe it is described on TEM.

4. Line 138: Please indicate the molar absorbance coefficient of RhB.

5. Line 180: The nano-alumina is very nice shape and dispersity. Please indicate DLS data for the nano-alumina and SMNA.

6. Line 242: Please indicate temperature during measurements.

Author Response

The authors have reported preparation of anionic sodium dodecyl sulfate (SDS) modified nano-alumina (SMNA) and its adsorption property of Rhodamine B (RhB). Nano-alumina was characterized using X-ray diffraction, Fourier transform infrared spectroscopy, Transmission Electron Microscopy, and BET methods. The optimum adsorption conditions were studied. The experiments were well organized, and the manuscript is well written. Although I believe there is a merit for readers and publisher to publish of the manuscript in Materials.

Thank you very much for your positive comments, by which we could improve the manuscript. The manuscript was revised by following your comments. All the changes and/or addition are in Red text.  My responses to each comment are below.

I feel that some parts should be improved. Please consider the following comments:

 1. Line 76: Please indicate purity of RhB.

The impurity of RhB (greater than 95 %) was added in the revised manuscript.

 2. Line 80: Methylene blue was used in this study. However one cannot find out the results of methylene blue.

Methylene blue was used as cationic dye to make ion pair formation with anionic surfactant SDS to determine concentration of SDS. By the way, section 3.2 (Modification of synthesized nano-alumina by SDS adsorption) is the results with the important role of Methylene blue.

 3. Line 116: Please indicate what kind of measurement. Maybe it is described on TEM.

Thank you very much for this point. TEM is the measurement. We added the measurement in the sentence.

The particle size distribution of synthesized nano-alumina which was evaluated by TEM, was performed by using Hitachi (H7650, Tokyo, Japan) with Olympus camera (Veleta 2kx2k).

 4. Line 138: Please indicate the molar absorbance coefficient of RhB.

The molar absorbance coefficient of RhB  determined by experiment is 105456 ±734 cm-1.M-1 that is very good agreement with the value of 106,0-1.M-1 for standard RhB.

This information was added in the revised manuscript.

 5. Line 180: The nano-alumina is very nice shape and dispersity. Please indicate DLS data for the nano-alumina and SMNA.

We added the information of DLS data as well as the detail of this experiments for the nano-alumina and SMNA.

Dynamic Light Scattering (DLS) for particle size characterization in the solutions of nano-alumina and SMNA was also conducted with Zetasizer Nano ZS by applying backscattering detection (173° detection optics) at 22°C. A cleaned plastic cuvette containing 2mL of the sample were used in each DLS measurement.

The DLS data (not shown in detail) indicates that the hydrodynamic Z-average diameter of nano-alumina was about 400–450nm at pH 4 that much higher than the particle size of nano-alumina due to aggregation of nano-alumina in NaCl concentration. The Z-average diameter of SMNA was about 924 -958 nm, indicating that the formation of SMNA aggregates in the presence of SDS. However, the aggregation of SMNA did not induce significantly to RhB adsorption.

 6. Line 242: Please indicate temperature during measurements.

We add the temperature of 25 ± 2 oC during the measurements in the revised manuscript.

Figure 8. The effect of contact time on RhB removal using synthesized nano γ-Al2O3 without SDS modification (temperature 25 ± 2 oC ,Ci (RhB) = 10-6 M and with SDS modification (Ci (RhB) = 10-4 M).

We also use the English editing service by MDPI to improve English of the paper. We hope the revised manuscript can be accepted after revising according to your comments.

Reviewer 2 Report

The authors have synthesized the alumina nanoparticles for absorbent of rhudamine, cationic dye. Similar paper titled, "Synthesis of g-alumina (Al2O3) nanoparticles and their potential for use as an adsorbent in the removal of methylene blue dye from industrial wastewater" by Shafqat Ali, Yasir Abbas, Zareen Zuhra and Ian S. Butler published in Nanoscale advances uses alumina nanoparticles for synthesizing methylene blue which is also a cationic dye like rhudamine. Please comment how is your work different from this? 

Please correct the value to +2.85 in line 292.

Author Response

The authors have synthesized the alumina nanoparticles for absorbent of rhudamine, cationic dye. Similar paper titled, "Synthesis of g-alumina (Al2O3) nanoparticles and their potential for use as an adsorbent in the removal of methylene blue dye from industrial wastewater" by Shafqat Ali, Yasir Abbas, Zareen Zuhra and Ian S. Butler published in Nanoscale advances uses alumina nanoparticles for synthesizing methylene blue which is also a cationic dye like rhudamine. Please comment how is your work different from this? 

Thank you very much for your comments, we added a paragraph to compare and discuss our work and your suggested paper. Also, the paper was added in the References.

In order to emphasize the high performance of SMNA for dye removal, let us discuss the potential in term of removal of cationic dye, methylene blue (MTB). The previously published paper by Ali el al. [51] successfully synthesized nano γ-Al2O3 by a precipitant method in the presence of nonionic surfactant Tween-80. This procedure required the ultrasonication and washing with ethanol before calcination at 5500C. In our case, the nano γ-Al2O3 was easily synthesized by titrating NaOH into Al(NO3)3 without any other chemicals. The nano γ-Al2O3 in this study achieved better morphology and much higher specific surface area than that in the paper [51]. For cationic dye adsorption, MTB was removed with removal efficiency of 98% while the removal efficiency of RhB using SMNA in this paper reached to 100% for the similar initial concentration of cationic dye. It implies that nano γ-Al2O3  is an  excellent adsorbent for MTB removal while SMNA is a novel adsorbent for RhB removal.

51. Ali S, Abbas Y, Zuhra Z, Butler IS. Synthesis of γ-alumina (Al2O3) nanoparticles and their potential for use as an adsorbent in the removal of methylene blue dye from industrial wastewater. Nanoscale Advances 2019 1 (1):213-21.

Please correct the value to +2.85 in line 292.

Thank you for this point. The value was corrected in the revised manuscript.

We also use the English editing service by MDPI to improve English of the paper. We hope the revised manuscript can be accepted after revising according to your comments.

Round  2

Reviewer 2 Report

Thank you for the revisions.